# Shikonin Attenuates Hepatic Steatosis by Enhancing Beta Oxidation and Energy Expenditure via AMPK Activation

**DOI:** 10.3390/nu12041133

**Published:** 2020-04-17

**Authors:** So Young Gwon, Jiyun Ahn, Chang Hwa Jung, BoKyung Moon, Tae-Youl Ha

**Affiliations:** 1Division of Food Functionality Research, Korea Food Research Institute, Wanju 55365, Korea; sygwon@foodinfo.or.kr (S.Y.G.); jyan@kfri.re.kr (J.A.); chjung@kfri.re.kr (C.H.J.); 2Department of Law Policy Research, National Food Safety Information Service, Seoul 110-750, Korea; 3Division of Food Biotechnology, University of Science & Technology, Daejeon 305-350, Korea; 4Department of Food and Nutrition, Chung-Ang University, Anseong 456-756, Korea; bkmoon@cau.ac.kr

**Keywords:** AMPK, fatty acid oxidation, hepatic lipid accumulation, shikonin, Hepa 1-6 cells, high fat fed mice

## Abstract

Shikonin, a natural plant pigment, is known to have anti-obesity activity and to improve insulin sensitivity. This study aimed to examine the effect of shikonin on hepatic steatosis, focusing on the AMP-activated protein kinase (AMPK) and energy expenditure in Hepa 1-6 cells and in high-fat fed mice. Shikonin increased AMPK phosphorylation in a dose- and time-dependent manner, and inhibition of AMPK with compound C inhibited this activation. In an oleic acid-induced steatosis model in hepatocytes, shikonin suppressed oleic acid-induced lipid accumulation, increased AMPK phosphorylation, suppressed the expression of lipogenic genes, and stimulated fatty acid oxidation-related genes. Shikonin administration for four weeks decreased body weight gain and the accumulation of lipid droplets in the liver of high-fat fed mice. Furthermore, shikonin promoted energy expenditure by activating fatty acid oxidation. In addition, shikonin increased the expression of PPARγ coactivator-1α (PGC-1α), carnitine palmitoyltransferase-1 (CPT1) and other mitochondrial function-related genes. These results suggest that shikonin attenuated a high fat diet-induced nonalcoholic fatty liver disease by stimulating fatty acid oxidation and energy expenditure via AMPK activation.

## 1. Introduction

Non-alcoholic fatty liver disease is a common chronic liver disease that is commonly associated with metabolic syndromes, such as obesity, type 2 diabetes, and hyperlipidemia [1]. Obesity is linked to significantly increased hepatic steatosis, induced by excessive fat accumulation [2]. The liver is a crucial organ that is responsible for whole body metabolism and homeostasis. Excessive hepatic lipid accumulation can trigger inflammation and more serious liver disorders [3]. Pro-inflammatory cytokines and lipogenic factors play crucial roles in the pathogenesis of liver steatosis, necrosis, and fibrosis [4].

AMP-activated protein kinase (AMPK) plays an important role in lipid metabolism and controlling metabolic disorders. AMPK is activated by Liver Kinase B1 (LKB1) under conditions of energy deficiency and protects the cell against ATP depletion by stimulating fatty acid oxidation and inhibiting lipid synthesis [5]. AMPK inhibits de novo fatty acid synthesis by inactivating acetyl-CoA carboxylase (ACC) and stimulates fatty acid oxidation by the upregulation of carnitine palmitoyltransferase-1 (CPT1) and peroxisome proliferator-activated receptor α (PPAR-α) [6]. Recent studies suggest that AMPK could be a therapeutic target for chronic liver disease treatment based on the activity of compounds such as glabridin [7], lipoic acid [8], and catechin [9]. The discovery of food components that can ameliorate hepatic lipid accumulation is therefore of interest. Metformin, a common antidiabetic drug, decreases hepatic steatosis by activating AMPK [10,11,12].

Shikonin is a naturally occurring naphthoquinone pigment and one of main active components present in the root of plant species *Lithospermum erythrorhizon* Siebold and Zucc (LE). LE has been used in vegetable side dishes and in the traditional Korean distilled liquor Jindo Hongju. Additionally, it has been used to treat a variety of disorders including macular eruptions, measles, sore throat, carbuncles, and burns [13]. Previous studies have demonstrated that shikonin exhibits anti-inflammatory [14] and anti-cancer effects [15,16]. Many studies have shown that shikonin can exert protective effects against obesity by modulating glucose tolerance, lipogenesis and β-oxidation [17,18,19]. Recent studies have shown that shikonin plays a significant role on AMPK activation against adipogenesis, diabetes, hepatic carcinoma and hepatic fibrosis [20,21,22,23]. Weijia Yang et al. reported that shikonin ameliorated hepatic lipid dysregulation through PPARγ and the MMP-9/TIMP-1 axis [24]. In addition, the naphthoquinone derivative of β-hydroxyisovalerylshikonin inhibited adipogenesis of 3T3-L1 cells through increased phosphorylation of AMPK and precursor SREBP-1c [25]. Therefore, shikonin exhibits several biological and pharmacological properties. However, much remains to be elucidated about the role of shikonin on hepatic lipid metabolism and nonalcoholic fatty liver disease (NAFLD).

In this study, we investigated the effects of shikonin on AMPK activation in murine Hepa 1-6 cells because AMPK has an important role in fatty acid oxidation. To clarify the relationship between the regulation of energy expenditure and AMPK activation by shikonin, we examined the effect of oral administration of shikonin on the AMPK phosphorylation and oxygen consumption rate in diet-induced obese mice. 

## 2. Materials and Methods 

### 2.1. Cell Culture 

Hepa 1-6 cells obtained from American Type Culture Collection were cultured in DMEM containing 10% fetal bovine serum, 100 U/mL penicillin, 100 µg/mL streptomycin, and 2 mM L-glutamine (Invitrogen, Carlsbad, CA, USA) at 37 °C under 5% CO_2_. Cells were grown to 90% confluence and then treated with various concentrations of shikonin (Sigma-Aldrich, St Louis, MO, USA) and incubated for various durations. To evaluate the effects of shikonin on lipid accumulation, the cells were treated with 100 μM oleic acid (Sigma-Aldrich, St Louis, MO, USA). After 24 h of incubation with oleic acid, protein and gene expression levels were evaluated as described below.

### 2.2. MTT Assay

Cell viability was determined using the MTT assay (Calbiochem, San Diego, CA, USA) in 96-well plates. Hepa 1-6 cells were seeded at a density of 1×10^4^ cells per well. After 48 h incubation, the cells were treated with 5 mg/mL MTT at 37 °C for 4 h. The reduction product, MTT-formazan, was solubilized with DMSO. The absorption at 570 nm was used as a measure of the MTT-reducing activity of the cells.

### 2.3. Oil Red O Staining

After the 24 h incubation with oleic acid, the cells were stained with Oil Red O (0.2% Oil Red O in 60% isopropanol). The cells were washed twice with PBS, fixed with 10% formalin for 1 h, dried, and stained with Oil Red O for 10 min. The cells were then washed with 70% ethanol and water and then dried. The lipid content of the stained cells was visualized by microscopy (Olympus IX71, Tokyo, Japan). 

### 2.4. Nile Red and DAPI Staining

After the 24 h incubation with oleic acid, the intracellular lipid droplets were measured by Nile Red staining. The cells were washed twice with PBS, fixed with 3% formalin for 20 min, and then ice-cold methanol for 10 min. The cells were then stained with 1 μg/mL Nile Red (Sigma-Aldrich) in PBS for 20 min at 37 °C and nuclei were counterstained with DAPI (Molecular Probes, Eugene, OR). The lipid content of the stained cells was visualized by fluorescence microscopy (Olympus IX71). Nile Red content levels were quantified by measuring fluorescence in a plate reader (TECAN infinite 200, excitation: 480 nm; emission: 580 nm). 

### 2.5. Protein Extraction and Western Blot Analysis

For Western blot analysis, the cells were washed with ice-cold PBS, and centrifuged. The harvested cells were sonicated for five seconds at 40 W. Cell lysates were incubated for 20 to 30 min on ice and then centrifuged at 13,000× *g* at 4 °C for 10 min. The protein concentration of the supernatant was determined using the Bio-Rad Protein Assay Reagent (Bio-Rad Laboratories, Hercules, CA, USA) using bovine serum albumin as the standard. The total protein sample (30 μg per lane) was separated by 10% SDS-polyacrylamide gel electrophoresis and transferred to polyvinylidene difluoride (PVDF) membranes (Millipore, Billerica, MA, USA). The membranes were blocked for 2 h at room temperature with 0.1% Tween 20 (Amresco Inc., Solon, OH, USA) in Tris-buffered saline containing 5% skim milk. After an overnight incubation at 4 °C with the specific primary antibodies, the membranes were incubated with a horseradish peroxidase-conjugated secondary antibody for 1 h at room temperature. Immunodetection was performed with the ECL detection reagent (Amersham Biosciences, Uppsala, Sweden). All figures showing the quantitative analyses include data from at least three independent experiments. 

### 2.6. RNA Extraction and Real-Time Quantitative RT PCR

Total RNA was extracted using the RNase kit (Nucleospin, iNtRON Biotechnology, Seongnam, Korea) and used to synthesize cDNA for quantitative real-time reverse transcription-polymerase chain reaction (qRT-PCR) analysis (StepOne Plus, Applied Biosystems, Carlsbad, USA). The qRT-PCR was performed in a 20 μL reaction mixture. The cycle conditions were as follows: 95 °C for 5 min followed by 50 cycles of denaturation at 95 °C for 20 s, annealing at 55 °C for 15 s, and extension at 72 °C for 30 s. The primer sequences were as follows: *PPARα* forward, 5′-AGAGCCCCATCTGTCCTCTC-3′; *PPARα* reverse, 5′- ACTGGTAGTCTGCAAAACCAAA-3′; *CPT1* forward, 5′-CTCCGCCTGAGCCATGAAG-3′; *CPT1* reverse, 5′-CACCAGTGATGATGCCATTCT -3′; *PGC-1α* forward, 5′-TGCAGC CAAGACTCTGTATG-3′; *PGC-1α* reverse, 5′-CATCAAGTTCAGAAAGGTCAAG-3′; *AMPKα* forward 5′-GTCAAAGCCGACCCAATGATA-3′; and *AMPKα* reverse, 5′- CGTACACGCAAATAATAGGGGTT-3′; *NRF1* forward, 5’- CCACGTTGGATGAGTACACG-3’; *NRF1* reverse, 5’- CAGACTCGAGGTCTTCCAGG-3’; *UCP2* forward, ‘5- AATGTTGCCCGTAATGCC-3’; *UCP2* reverse, 5’- CCCAAGCGGAGAAAGGAA-3’.

### 2.7. Oxygen Consumption Rate

We measured cellular respiration with a Seahorse Biotechnology XF24 extracellular flux analyzer (Seahorse Bioscience, Billerica, MA) [3]. Hepa 1-6 cells were grown in Seahorse XF 24-well plates. We treated the cells with shikonin for 24 h. Cells were equilibrated to the unbuffered medium for 1 h at 37 °C in a CO_2_-free incubator, before being transferred to the XF24 analyzer. We measured the basal oxygen consumption rate (OCR), and then injected 0.2 mM palmitic acid in BSA or BSA vehicle alone. After 3–5 min of mixing, multiple OCR measurements were made. 

### 2.8. High-Fat Fed Mouse Model

C57BL/6 male mice were obtained from Oriental Bio Inc (Gyeonggi-do, Korea). Each group containing 20 mice were housed in cages at 40–60% humidity at a controlled temperature (20–26 °C), with a 12 h light/dark cycle. Mice had free access to water and the experimental diet which was based on the AIN-76 diet. The high-fat diet (HFD) contained 25% fat (lard 200 g/kg, corn oil 50 g/kg) and 0.5% cholesterol (w/w). Mice were fed a HFD for eight weeks to induce obesity. After this, the mice were split into two groups: HFD and HFD+S. The HFD group (vehicle control) was dosed by oral gavage with soybean oil. Shikonin (30 mg/kg/day) dissolved in soybean oil was administered by oral feeding for 4 weeks to the HFD+S group. At the end of the experimental period, the mice were fasted for 12 h, and then sacrificed. All animal procedures were approved by the Korea Food Research Institutional Animal Care and Use Committee (KFRI 14005).

### 2.9. Histological Analysis

To examine the histological changes, the white adipose tissue (WAT) and livers were fixed in 10% formalin for one day and processed in a routine manner for paraffin sectioning. Five-micrometer-thick sections were cut and stained with hematoxylin and eosin (H&E) for microscopic examination (Leica RM2235, Wetzlar, Germany). Images were collected on a microscope (Olympus BX51, Tokyo, Japan). 

### 2.10. RQ, VO_2_ Max, and Energy Expenditure Measurements

Whole-body oxygen consumption (VO_2_) and carbon dioxide production (VCO_2_) were measured using a LE 405 gas analyzer (Panlab Harvard Apparatus, Barcelona, Spain). Before the experiment, gas calibration was performed with a mixture of 5% CO_2_, 50% O_2_, and 20% O_2_ gas standards. VO_2_ and VCO_2_ were measured at three-minute intervals during the dark phase for 8 h. The cages were provided with a constant airflow rate of 2 L/min. A respiratory quotient (RQ) was calculated from the VO_2_ and VCO_2_ values. Energy expenditure was calculated as the product of the oxygen caloric value and VO_2_ per kilogram of body weight, according to Wire’s equation [26].

### 2.11. Statistical Analysis

The group results were compared by an analysis of variance (ANOVA), followed by Tukey’s HSD test using SPSS 18.0 software (IBM, Armonk, NY, USA). The data are expressed as the mean ± standard error of the mean (SEM). *p* < 0.05 was considered significant.

## 3. Results

### 3.1. Shikonin Increases AMPKα Phosphorylation in Hepa 1-6 Cells

The cytotoxicity of shikonin was measured using the MTT assay after 48 h of treatment. As Figure 1A shows, shikonin was not cytotoxic in hepatocytes at a concentration range of 0.5–2 μM. Next, the possible role of shikonin on AMPKα and downstream acetyl CoA carboxylase (ACC) phosphorylation was examined by Western blot analysis. Shikonin significantly increased AMPKα and ACC phosphorylation in a dose-dependent manner (Figure 1B). Furthermore, AMPKα and ACC phosphorylation increased over time and peaked 120 min after shikonin treatment (Figure 1C). However, LKB1 levels did not change following shikonin treatment. These results demonstrated that shikonin activates AMPKα phosphorylation.

We next examined whether shikonin directly increased AMPK activation using compound C and metformin treatment. Metformin, an AMPK activator, also increased AMPK phosphorylation in Hepa 1-6 cells, and these effects were inhibited by compound C, an AMPK inhibitor. The results showed that shikonin treatment partly recovered AMPK phosphorylation reduced by compound C in Hepa 1-6 cells (Figure 1D). These results show that shikonin increases AMPK phosphorylation in hepatocytes.

### 3.2. Shikonin Attenuates Lipid Accumulation in Hepa 1-6 Cells

To investigate the effects of shikonin on lipid accumulation in hepatocytes, Hepa 1-6 cells were incubated with oleic acid only (100 μM) or in combination with shikonin for 24 h. After incubation, intracellular lipid accumulation was determined by Oil Red O and Nile Red staining. Oil Red O and Nile Red staining revealed that intracellular lipid content was reduced by shikonin treatment (Figure 2A). As shown in Figure 2B, fluorescence intensity levels were significantly reduced in cells treated with 2 μM shikonin compared to oleic acid alone.

To further elucidate the molecular mechanism underlying the inhibitory effects of shikonin against lipid accumulation, we determined the protein levels of sterol regulatory element binding protein-1c (SREBP1c) and AMPKα. Shikonin treatment decreased levels of SREBP1c and increased phosphorylation levels of AMPK and ACC (Figure 3A). Moreover, the oleic acid-induced increase in AMPK phosphorylation was reduced in the presence of compound C (Figure 3B). These data suggest that shikonin is involved in AMPK activation.

### 3.3. Shikonin Enhances Fatty Acid Oxidation in Hepa 1-6 Cells

AMPK activation inhibits ACC activity, leading to a reduction in malonyl-CoA and an increase in CPT1 activity. One expected consequence of this is increased fatty acid oxidation [27].

We next evaluated the effects of shikonin on hepatic fatty acid oxidation. The oxidation of fatty acids can be assessed by monitoring cellular oxygen consumption [28]. Hepa 1-6 cells were pretreated for 24 h with shikonin. After treatment for 24 h, O_2_ consumption was measured using an XF 24 analyzer. Shikonin treated cells showed significantly enhanced fatty acid oxidation compared to the control cells (Figure 4A). In parallel with increased fatty acid oxidation by shikonin, the mRNA expression of genes involved in fatty acid oxidation such as *PPARα, CPT1*, and *PGC-1α* was increased (Figure 4B). These results suggest that shikonin can protect against excessive lipid accumulation through fatty acid oxidation in Hepa 1-6 cells.

### 3.4. Shikonin Ameliorated High-Fat Diet-Induced Obesity

To examine the effect of shikonin on the developed obesity, mice were fed a high-fat diet for eight weeks to induce weight gain. After, the mice were assigned to one of two groups: HFD and HFD+S. The HFD+S mice were provided shikonin (30 mg/kg/day) in oral administration for 4 weeks. During the experimental period, we measured the body weights of mice once a week and found that shikonin significantly decreased the body weight gain (Figure 5). Additionally, shikonin significantly reduced the weight of WAT (both epididymal and retroperitoneal WAT) as well as adipocyte size. 

### 3.5. Shikonin Increases Energy Expenditure

To investigate whether shikonin affects energy expenditure, we measured oxygen consumption by indirect calorimetry. Whole-body O_2_ consumption (VO_2_) was significantly higher in HFD+S mice compared to HFD mice. In addition, shikonin administration significantly increased VCO_2_ (Figure 6A). Furthermore, the RQ, which reflects the ratio of carbohydrate to fatty acid oxidation, was significantly lower in HFD+S mice, indicating that these mice used a greater portion of fatty acids as a fuel source for energy production than the HFD mice. As shown in Figure 6D, HFD+S mice also exhibited increased energy expenditure (Figure 6D). These data are consistent with the increased oxygen consumption rate observed in Hepa 1-6 hepatocytes. 

### 3.6. Shikonin Prevents High-Fat Diet-Induced Hepatic Steatosis

Mice fed a high-fat diet for several weeks accumulate lipids in their livers and eventually develop liver steatosis [29]. However, in liver sections stained with H&E to visualize lipid content, we found a decrease lipid accumulation in HFD+S mice (Figure 7A). Moreover, shikonin reduced hepatic total lipid, TG, and total cholesterol levels that were elevated in HFD mice (Figure 7B–D).

Next, we evaluated the effect of shikonin on AMPK activation associated with energy metabolism. In agreement with our in vitro results, oral administration of shikonin increased the phosphorylation of AMPK in liver tissue. In addition to the changes in liver tissue, shikonin increased phosphorylation of AMPK in the skeletal muscle of HFD+S mice compared to the control group (Figure 7E). We also demonstrated that shikonin increased the mRNA expression of genes involved in fatty acid oxidation such as *PPARα, CPT1*, and *PGC-1α* (Figure 7F). Shikonin significantly increased mRNA expressions of *UCP2*. These findings demonstrate that shikonin-induced AMPK activation increases fatty acid oxidation in the liver and muscles.

## 4. Discussion

In the present study, we investigated the effect of shikonin and its action mechanisms on obesity-induced hepatic steatosis by using OA-treated Hepa 1-6 cells and HFD-induced obese mice models. Shikonin effectively reduced the level of lipid accumulation in Hepa 1-6 cell and increased the phosphorylation levels of AMPK and ACC. Furthermore, shikonin enhanced cellular oxygen consumption and significantly increased the fatty acid oxidation related genes such as PPARα, CPT1, and PGC-1α. More importantly, shikonin significantly increased whole-body O_2_ consumption in obese mice. Shikonin is the main component of LE roots, which have been reported to provide beneficial health effects to various conditions such as inflammation and cancer [13]. LE roots were originally used as colorants for foods such as Jindo Hongju (Korea traditional liquor). In addition, shikonin has been reported to inhibit fat accumulation in 3T3-L1 cells [19] and to prevent high-fat diet-induced obesity in vivo [28,30]. Several studies have reported that shikonin can effectively ameliorate obesity and hepatic fibrosis [23,24,25]. We previously demonstrated the anti-obesity effect of shikonin by inhibiting adipogenesis and lipogenesis and increasing β-oxidation [18,19]. Shikonin also attenuated liver fibrosis by downregulating the transforming growth factor-β1/Smads pathway and by inhibiting autophagy [31]. Here, we have focused on the effects of shikonin on fatty liver disease and AMPK activation. We observed that shikonin might have a role as a novel activator of AMPK, and by this pathway it may ameliorate high-fat diet-induced fatty liver disease.

AMPK is an energy regulator that plays a major role in glucose and lipid metabolism [32]. Activation of AMPK is closely correlated with energy metabolism in organs such as the liver, skeletal muscles, and adipose tissue [33]. An important finding in this study is that shikonin increased the phosphorylation of AMPKα in hepatocytes. We observed dose- and time-dependent effects of shikonin on stimulation of AMPKα phosphorylation in Hepa 1-6 cells (Figure 2). Shikonin gradually increased AMPKα phosphorylation in Hepa 1-6 cells, which peaked after 120 min. To further validate our results, we examined using compound C, a chemical inhibitor of AMPK. AMPK phosphorylation inhibited by compound C was partly reversed by shikonin treatment. Although shikonin increased AMPKα phosphorylation, the LKB1 level did not change. This is notable because AMPK is typically activated through phosphorylation by the upstream kinase LKB1 [34]. However, it has also been reported that AMPK is activated through reversible phosphorylation by LKB1 [35]. AICAR is an adenosine analog that directly modulates AMPK with [32] or without direct activation of LKB1 [36]. Thus, it is speculated that other possible AMPK candidates may be involved in the phosphorylation of AMPK by shikonin.

Previous studies have reported that activation of AMPK results in decreased hepatic triglyceride and enhanced fatty acid oxidation [37,38]. We investigated the effects of shikonin treatment on oleic acid-induced steatotic hepatocytes. This study showed that shikonin significantly decreased oleic acid-induced lipid accumulation in Hepa 1-6 cells (Figure 3). Activation of AMPK reduces expression of the lipogenic transcription factors SREBP1 and ACC [29]. ACC is a key enzyme for fatty acid synthesis and inactivation of ACC by AMPK phosphorylation leads to a decrease in fatty acid synthesis and stimulates β-oxidation [39,40]. We also found that shikonin treatment increased phosphorylation of AMPK and ACC, a direct target of AMPK, in oleic acid-treated cells. Additionally, sterol regulatory element binding protein-1c (SREBP1c) expression was reduced by shikonin treatment. AMPKs downregulate SREBP1c, which is a key transcription factor regulating de novo lipogenesis, in turn decreasing fatty acid synthesis [41,42]. It has been demonstrated that activation of AMPK leads to the regulation of several downstream targets involved in lipid metabolism [43]. Our results indicate that shikonin activates AMPK which suppressed fatty acid synthesis. 

In addition to the increased AMPK activation, we showed a significant increase in cellular oxygen consumption rate and the expression of fatty acid oxidation related genes such as *PPARα*, *CPT1*, and *PGC-1α* (Figure 5). PPARα is one of the nuclear receptor families that play a crucial role in lipid homeostasis [44]. PPARα coordinates the transcriptional activation of carnitine palmitoyl transferase-1 (*CPT1*), leading to fatty acid oxidation [45]. Furthermore, PPARγ coactivator-1α (PGC-1α) cooperates with PPARα and induces mitochondrial fatty acid oxidation [46]. 

Shikonin-induced phosphorylation of AMPK in hepatocytes was also confirmed in vivo. Shikonin increased the phosphorylation of AMPKα in the liver and muscles of diet-induced obese mice (Figure 7E). The oral administration of shikonin also increased energy expenditure in obese mice (Figure 6). Furthermore, histological analysis showed that oral administration of shikonin can ameliorate hepatic steatosis in mice fed a high-fat diet. Notably, shikonin significantly increased mRNA levels of PGC-1α both in vivo and in vitro model. AMPK could directly activate PGC-1α and regulate the promoter of PGC-1α [47]. PGC-1α plays a crucial role in the regulation of mitochondrial biogenesis by coordinating the activity NRFs, UCP2 and CPT1, which are genes responsible for mitochondrial fat oxidation metabolism [48,49]. Thus, this protective effect of shikonin might be linked to the activation of AMPK to increase hepatic β-oxidation, leading to an increase in energy expenditure as well as a decrease in lipid levels in the liver [37]. Based on our study, the effect of shikonin on lipid metabolism is mediated by the activation of the AMPK pathway. 

In conclusion, shikonin plays a significant role in reducing hepatic lipid accumulation via the activation of AMPK which enhances β-oxidation and energy expenditure, protecting against hepatic steatosis in mice fed a high-fat diet. Based on these results, shikonin possesses therapeutic potential for preventing non-alcoholic fatty liver disease. Further studies are needed to establish a direct link between shikonin and AMPK using AMPK KO mice, as well as models for oxidative stress in the mitochondrial respiratory chain and chronic inflammation, to better understand the protective effect of shikonin on non-alcoholic fatty liver.

## Figures and Tables

**Figure 1 nutrients-12-01133-f001:**
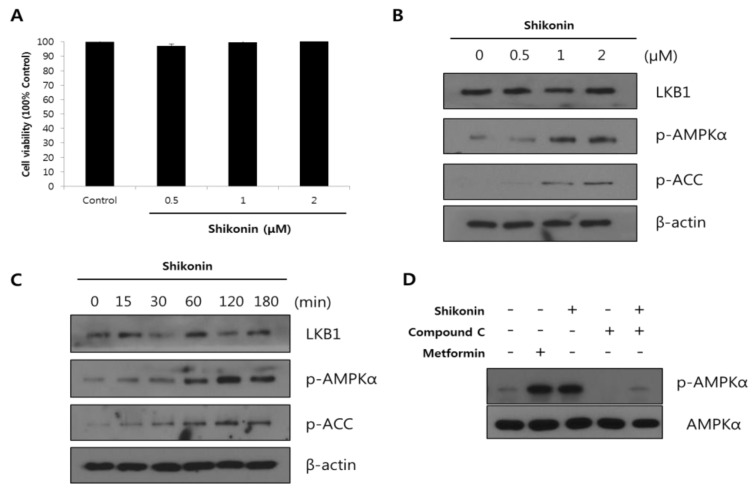
Shikonin stimulates the phosphorylation of AMP-activated protein kinase α (AMPK) and acetyl-CoA carboxylase (ACC) in Hepa 1-6 cells. Cell cytotoxicity was determined by MTT assay (**A**). Hepa 1-6 cells were treated with the indicated concentrations of shikonin for 1 h (**B**) or with 2 μM shikonin for various durations (**C**). Effect of AMPK inhibitor on phosphorylation of AMPKα in Hepa 1-6 cells treated with shikonin (**D**). Cells were treated with either compound C (40 μM) or metformin (2 mM) alone for 2 h, or exposed to 40 μM compound C for 1 h, then were treated with 2 μM shikonin for 2 h.

**Figure 2 nutrients-12-01133-f002:**
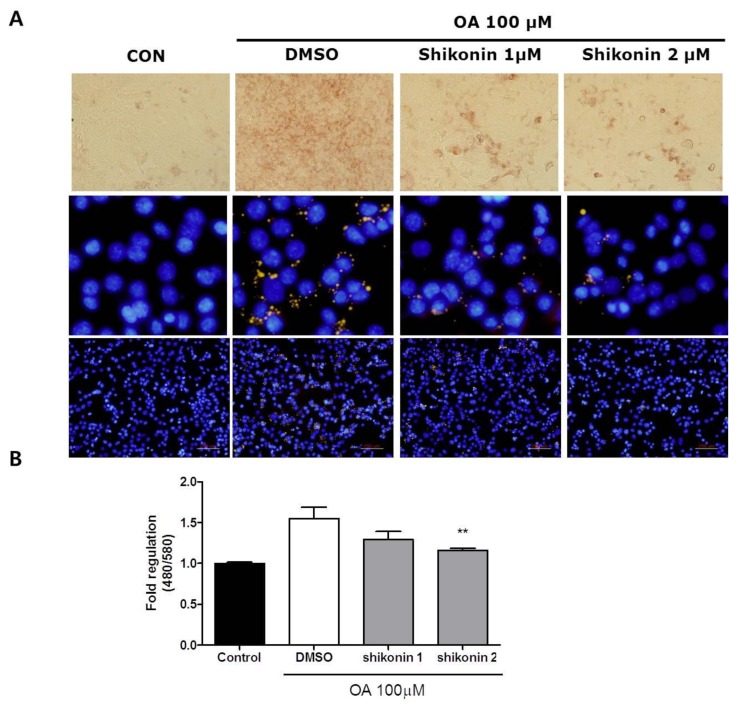
Shikonin inhibits oleic acid-induced lipid accumulation in Hepa 1-6 cells. (**A**) The lipid content was assessed by Oil Red O staining and Nile Red staining. (**B**) Quantitative assessment was measured by flow cytometry. The result was expressed as mean ± standard error of the mean (SEM). **, *p* < 0.05.

**Figure 3 nutrients-12-01133-f003:**
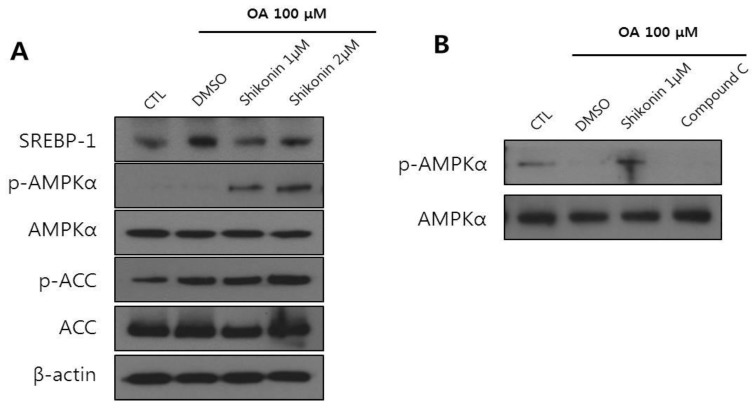
Shikonin increases AMPK phosphorylation and decreases sterol regulatory element binding protein-1c (SREBP-1c) in oleic acid-induced hepatic steatosis. (**A**) Hepa 1-6 cells were pretreated with 100 μM oleic acid and then incubated with shikonin for 24 h. (**B**) Cells were pretreated with 100 μM oleic acid and then incubated with either compound C (40 μM) or shikonin (2 μM) for 2 h. Protein expression was measured by Western blot analysis.

**Figure 4 nutrients-12-01133-f004:**
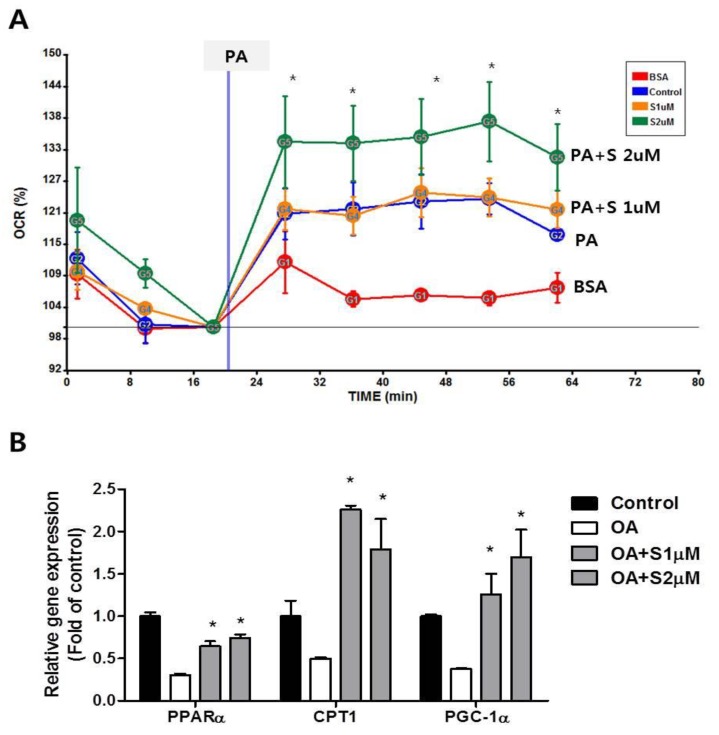
Shikonin stimulates fatty acid oxidation. (**A**) Hepa 1-6 cells were incubated with shikonin for 24 h, and fatty acid oxidation was measured as described in the experimental section. (**B**) Hepa 1-6 cells were pretreated with 100 μM oleic acid and then incubated with shikonin for 24 h. Relative mRNA expression levels of fatty acid oxidation related genes were measured by quantitative real-time reverse transcription-polymerase chain reaction (qRT-PCR). The results were expressed as mean ± SEM. *, *p* < 0.05.

**Figure 5 nutrients-12-01133-f005:**
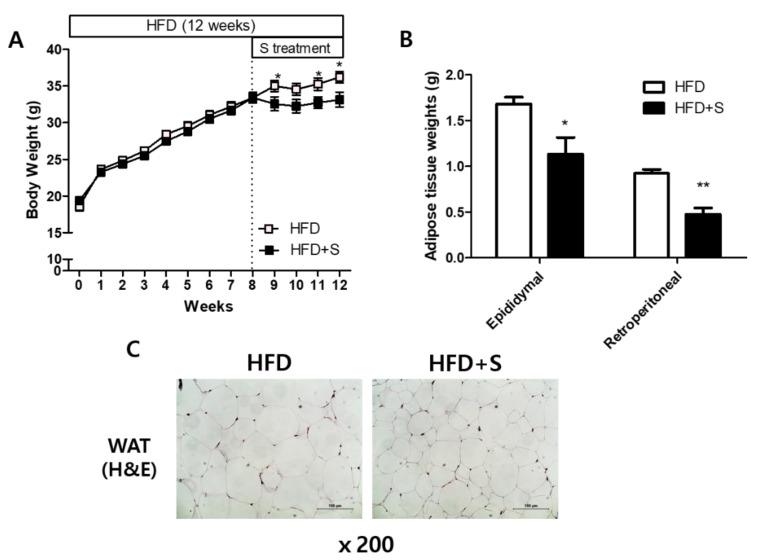
Shikonin reduces body weight and white adipose tissue (WAT) weight. (**A**) Body weight, (**B**) weights of epididymal WAT and retroperitoneal WAT, (**C**) epididymal fat tissue morphology in high fat diet (HFD)-fed mice. The results were expressed as mean ± SEM. *, *p* < 0.05.

**Figure 6 nutrients-12-01133-f006:**
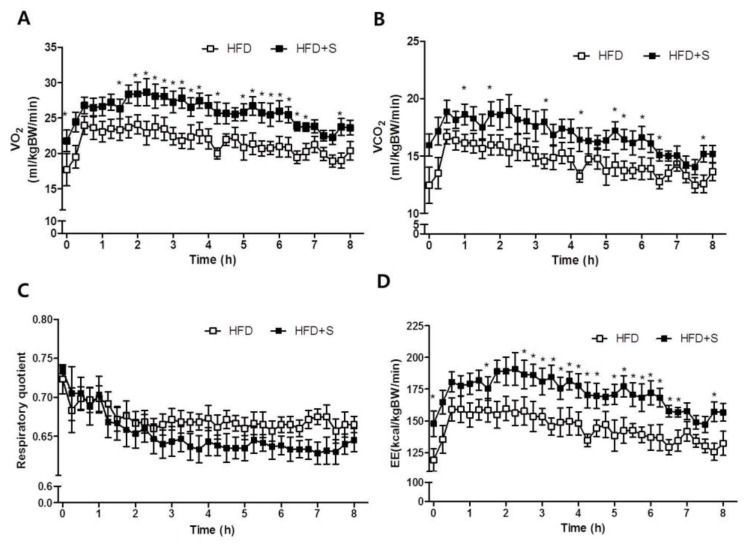
Shikonin stimulates fatty acid oxidation and promotes energy expenditure in HFD-induced obese mice. HFD-induced obese mice were orally administrated vehicle or shikonin for 4 weeks. After administration, mice were transferred to a chamber and (**A**) VO_2_, (**B**) VCO_2_, (**C**) respiratory quotient, and (**D**) energy expenditure were measured using indirect calorimetry for 8 h during dark cycles. Values represent the mean ± SEM of eight mice.

**Figure 7 nutrients-12-01133-f007:**
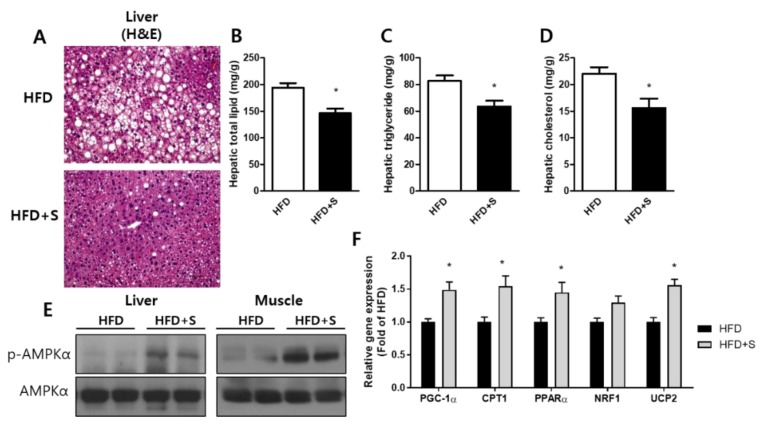
Shikonin ameliorates hepatic steatosis in HFD-induced obese mice. (**A**) Hepatic morphology changes were shown by hematoxylin and eosin (H&E) staining of liver sections. Original magnification, 200×. (**B**–**D**) Hepatic levels of total lipids, triglycerides, and total cholesterol were determined as described in the experimental section. (**E**) In liver and muscle total lysates, AMPK protein levels were determined by Western blot analysis. (**F**) In liver, relative mRNA expression levels of fatty acid oxidation and mitochondrial function related genes were measured by qRT-PCR. The results were expressed as mean ± SEM. *, *p* < 0.05.

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
