# Peer review of "Shikonin Attenuates Hepatic Steatosis by Enhancing Beta Oxidation and Energy Expenditure via AMPK Activation"

_nutrients, 2020, doi:10.3390/nu12041133_

Round 1

Reviewer 1 Report

This research article focuses on the effects of shikonin on hepatic steatosis and showed that shikonin increases fatty acid oxidation and phosphorylation of AMPK. The authors state that this research article is the first report to demonstrate that shikonin ameliorates hepatic steatosis through AMPK activation. However, these two effects of shikonin have already been demonstrated previously by others and none of these studies have been cited here, probably due to an outdated bibliography.

As search for new drugs is important in the context of non-alcoholic fatty liver diseases, this study would be interesting if the authors could demonstrate a strong link between shikonin mode of action, AMPK activity and modulation of lipid catabolism in hepatocytes.

Bibliography in general should be double-checked. A lot of citations are not correct – the citing papers do not demonstrate what is stated by the authors or in some cases literature should be “refreshed”.

  1. Missing literature:
  • The authors should use an up-to-date bibliography in order to introduce and discuss properly the already known roles of shikonin.

First, the authors cite two of their previous papers to introduce that « shikonin also inhibits adipogenesis in 3T3L1 (Lee H. 2011, Lee H. 2013) » line 55. However, more recent research articles showed that shikonin doesn’t only play a role in adipogenesis in vitro, but also in vivo in mice. More interestingly, some of these studies already reported an effect of shikonin on hepatic lipid metabolism:

  • Decreased Adiposity and Enhanced Glucose Tolerance in Shikonin Treated Mice. Ahmed Bettaieb et al., Obesity, 2015.
  • Shikonin protects against obesity through the modulation of adipogenesis, lipogenesis, and β-oxidation in vivo. So YoungGwon et al., Journal of functional foods, 2015.
  • Some papers also studied the effects of shikonin on hepatic steatosis, a novel contradiction with what is stated in the present study - Line 57 : «  the effects of shikonin on hepatic steatosis has not been studied ». The authors should modify this statement and cite :
  • Shikonin from Zicao prevents against non-alcoholic fatty liver disease induced by high-fat diet in rats. Weijia Yang et al., Academia Journal of Scientific Research, 2017

The authors should also discuss about the effects of Shikonin on hepatic fibrosis, another feature of nonalcoholic fatty liver diseases :

  • Shikonin alleviates hepatic fibrosis and autophagy via inhibition of TGF-β1/Smads pathway. Tong Liu et al., Journal of Gastroenterology and Hepatology, 2019

Effects of derivatives of shikonin on hepatic steatosis should also be pointed out :

  • Efficacy of Acetylshikonin in Preventing Obesity and Hepatic Steatosis in db/db Mice. Mei-Ling Su et al., 2016, Molecules.
  • Finally, others already showed that shikonin and its derivatives lead to an activation of AMPK in different cell types, including hepatocyte cell lines :
  • AMPK and SREBP-1c Mediate the Anti-Adipogenic Effect of β-Hydroxyisovalerylshikonin. Jong-Hyeok Ha et al., International Journal of Molecular Medicine, 2017.
  • A novel natural compound Shikonin inhibits YAP function by activating AMPK. Fang-Jie Yan et al., TMR Modern Herbal Medicine, 2018.
  • Shikonin exerts antitumor activity by causing mitochondrial dysfunction in hepatocellular carcinoma through PKM2-AMPK-PGC1α signaling pathway. Bing Liu et al., Biochemistry and Cell Biology, 2019.
  1. The direct link between shikonin, lipid catabolism and AMPK is not evaluated.

The authors use a siRNA approach in some of their figures to demonstrate that upon AMPK-knockdown shikonin does not increase AMPK phosphorylation. However, when lipid content or mitochondrial activity are assessed in vitro, this control is not present. In conclusion, it is impossible to evaluate that increase of AMPK phosphorylation by shikonin is responsible for the increase of fatty acid oxidation and decrease of lipid content in cells.

As shikonin was shown previously to modulate hepatic steatosis and AMPK phosphorylation, introduction of the siRNA-AMPK in some of the experiment would be beneficial for this study and add novelty.

To do so, the authors should evaluate for example if treatment of Hepa 1-6 with shikonin upon AMPK knockdown still :

  • decreases lipid droplets content (Fig 3A and B)
  • increases fatty acid oxidation (Fig 5A)
  • increases fatty acid oxidation related genes/PPARalpha target genes (Fig 5B)

The same statement could be made in vivo as the authors only describe two different effects of shikonin (decrease of hepatic steatosis and increase of AMPK phosphorylation) without demonstrating a mechanistic link. It is clear that the use of AMPK-knockout mice in this study would probably be beyond the scope of the manuscript, however the authors could use compound C, the pharmacological inhibitor of AMPK, in order to evaluate if co-treating mice with shikonin and compound C still lead to a decrease of hepatic steatosis.

  1. Literature is not always correct or some informations are missing.

Citation (5), (6) and (7) : the authors introduce with these citations that different drugs were shown to modulate AMPK activity and hepatic steatosis. But :

  • The authors should also discuss about Metformin, one of the well known drug increasing AMPK activity and currently used to modulate hepatic lipid metabolism.
  • Shikonin, as Metformin, seems to interfere with mitochondria functions. A review from Smith BK et al. (AJP) reported in 2016 that almost 30 studies examined the relationship of AMPK activation and fatty liver. Numerous drugs activating AMPK impact mitochondrial respiratory chain and it is thus important to evaluate or at least discuss about the role of shikonin on mitochondrial stress/viability. In this context, inflammation markers, liver enzymes, oxidative stress could be evaluated in vitro and in vivo.

Some other citations are not correct, among them :

Citation (4) this article doesn’t show that AMPK regulates CPT1 and PPARa expression.

Citation (5) is not talking about glabridin and not about treatment of fatty liver

Author Response

Response to Reviewer

We would like to thank Reviewer for insightful and constructive comments. These comments helped us strengthen and clarify our report. We have revised major and minor point in the manuscript in response to the suggestions of Reviewer as noted below.

Q1. Missing literature:

A1. We added new references in revised manuscript (line54-63, References).

Q2. The direct link between shikonin, lipid catabolism and AMPK is not evaluated.

A2. We agree with your opinions. Unfortunately, we did not confirm the direct link between shikonin and AMPK in this study. We are planning to confirm this point by using AMPK knockout mice in our next study and described this to revised text (line 333-336).

Q3. Literature is not always correct or some information are missing.

A3. As suggested, we changed revised manuscript (line 45-49). Also, we have corrected References as you pointed out (Changed reference).

Reviewer 2 Report

In this paper the researchers focused their attention on the role of shikonin, a plant derived compound, in the modulation of fat accumulation in an in vitro model of Hepa 1-6 cells and in high-fat fed mice. In this context they demonstrated a crucial role played by AMPK phosphorylation in determining an increase of fatty acid oxidation, directly linked to shikonin biological activity, both in cellular and animal model.

In my opinion the research project was well-designed. The methodology used is coherent with the aim of the study and the obtained results. In general, the paper is well-written, clear and concise.

  • Pay attention on the choice of keywords: in my opinion the use of “hepatic steatosis” and “non-alcoholic fatty liver disease” is redundant.
  • Line 32: please reformulate the sentence. The triglycerides accumulation in the hepatocyte not always is due to insulin resistance.
  • Minor English revision is necessary.
  • Please add at the end of the discussion the future perspectives. I also suggest you the evaluation of anti-inflammatory effects of shikonin in this context.
  • Silencing the activity of AMPK is it possible to observe a worsening of triglycerides accumulation in shikonin+oleic acid cell cultures? The section 3.2 could be improved thanks to this information.

Author Response

Response to Reviewer

We would like to thank Reviewer for insightful and constructive comments. These comments helped us strengthen and clarify our report. We have revised major and minor point in the manuscript in response to the suggestions of Reviewer as noted below.

Q1. Pay attention on the choice of keywords: in my opinion the use of “hepatic steatosis” and “non-alcoholic fatty liver disease” is redundant.

A1. Thank you for your comment. We have changed keywords. (line 29).

Q2. Line 32: please reformulate the sentence. The triglycerides accumulation in the hepatocyte not always is due to insulin resistance.

A2. We have corrected as you pointed out (line 37-39).

Q3. Minor English revision is necessary.

A3. According to your comment, our revised manuscript has been edited by native speaker (www. editige.co.kr).

Q4. Please add at the end of the discussion the future perspectives. I also suggest you the evaluation of anti-inflammatory effects of shikonin in this context.

A4. According to your suggestion, we described this point to the discussion part (line 333-336).

Round 2

Reviewer 1 Report

A few modifications in the text - as requested previously - have been done by the authors. These modifications are helpful to prevent some over-interpretations made by the authors. However, some problems still occur in the text. As said previously the use of AMPK-KO mice is clearly not expected in this study, but an additional and easy (as siRNA-AMPK was already used) in vitro experiment would have been really helpful to show the role of AMPK in shikonin-induced fatty acid oxidation and expression of fatty acid oxidation associated genes. Finally, It is really crucial not to omit the literature about shikonin and its derivatives in hepatic diseases.

Major:

1) Modifications for the siRNA and compound C :

As asked previously, authors have to be careful with the interpretation of the siRNA-AMPK and the use of compound C. Using siRNA-AMPK and compound C, only to evaluate the effect of shikonin on the phosphorylation of AMPK and ACC … is only useful to validate that the antibodies are effectively specific for AMPK, Phospho-AMPK and phospho-ACC. Knockdown by siRNA or inhibition with compound C will indeed logically decrease the expression/phosphorylation of AMPK. siRNA and compound C are already used in these cells and in presence of shikonin but not for the other in vitro experiments (mitochondrial activity, gene expressions), which doesn’t make sense if the authors want to really state that shikonin activates AMPK. Finally: “These results suggest that shikonin directly phosphorylates AMPK.” – is confusing, as shikonin certainly does not have a kinase activity.

These should be clarified:

Line 178: “The shikonin induced increase in AMPK phosphorylation reduced in the presence of compound C (Fig. 1D).”

Line 189-195: To confirm the critical role of shikonin-induced AMPK activation, we used a specific siRNA against the AMPK gene. AMPK protein and mRNA expression levels were significantly suppressed by the AMPK siRNA compared to the control siRNA (Fig. 2). Further, knockdown of AMPK abolished shikonin mediated AMPK activation. In addition, ACC phosphorylation was dramatically decreased by siRNA transfection. The role of shikonin in AMPK activation was verified by the lack of downstream ACC activation in Hepa 1-6 cells deficient of AMPK. This data suggests that shikonin directly regulates AMPK activation.

Line 297-301: To evaluate the involvement of shikonin on direct AMPK activation, we used chemical and genetic approaches to inhibit AMPK. Compound C disrupted the shikonin-induced phosphorylation of AMPK (Fig. 1C). In a similar approach, silencing AMPK expression using siRNA transfection suppressed the shikonin-induced phosphorylation of AMPK (Fig. 2). These results suggest that shikonin directly phosphorylates AMPK.

2) Same statement for in vivo experiments :

The authors should be careful about the fact that no link is shown here between AMPK and fatty acid oxidation/steatosis. The authors should reformulate some of these conclusions. For example : (please double-check the entire masnuscript)

Line 259: Shikonin prevents high-fat diet-induced hepatic steatosis via AMPK activation

Line 277-278: This study demonstrated that shikonin improves AMPK phosphorylation, increasing fatty acid oxidation in the liver, and consequently leading to a decrease in hepatic steatosis.

3) References in the text have been added or modified and helped to better understand what is already known about the effect of shikonin and derivatives. One major effect of shikonin have unfortunately been again omitted.

Line 283-284: However, the effect of shikonin on fatty liver disease had not been previously examined.

As asked previously, this should be cited and discussed:

Shikonin from Zicao prevents against non-alcoholic fatty liver disease induced by high-fat diet in rats. Weijia Yang et al., Academia Journal of Scientific Research, 2017

Shikonin alleviates hepatic fibrosis and autophagy via inhibition of TGF-β1/Smads pathway. Tong Liu et al., Journal of Gastroenterology and Hepatology, 2019

Efficacy of Acetylshikonin in Preventing Obesity and Hepatic Steatosis in db/db Mice. Mei-Ling Su et al., 2016, Molecules.

4) As reminded above, be careful with over-interpretations and conclusions.

LKB1 is indeed an activator of AMPK. However the authors only measure the protein expression of LKB1. It cannot unfortunately be used as a indicator of LKB1 activity as LKB1 can be phosphorylated etc. Moreover, as asked on the 1st version of the reviewing – and explaining why metformin should be added as another activator of AMPK – is that shikonin (as metformin) seems to have effects on the mitochondria itself. So to temper their interpretation and add more depth to the discussion, the authors should discuss about the possible mode of action of shikonin on AMPK.

For example:

  • LKB1 cannot be excluded here
  • Shikonin could target the mitochondria and modulate the redox state of the cell

In conclusion, this should be modified:

Line 293-296: "Although shikonin treatment increased AMPKα phosphorylation, the LKB1 level did not change. This is notable because AMPK is typically activated through phosphorylation by the upstream kinase LKB1. These results show that shikonin activates AMPK but did not affect the upstream kinase LKB1. Therefore, shikonin may directly be responsible for the activation of AMPK."

Minor:

It would be really interesting, if available, to add on fig. 6:

  • weight follow-up of mice during HFD and after Shikonin treatment
  • Physical activity of these mice, as increase of AMPK phosphorylation in the muscle could be linked to more activity and could potentially explain weight loss, decrease of hepatic steatosis …

Author Response

Major:

Q1) Modifications for the siRNA and compound C :

As asked previously, authors have to be careful with the interpretation of the siRNA-AMPK and the use of compound C. Using siRNA-AMPK and compound C, only to evaluate the effect of shikonin on the phosphorylation of AMPK and ACC … is only useful to validate that the antibodies are effectively specific for AMPK, Phospho-AMPK and phospho-ACC. Knockdown by siRNA or inhibition with compound C will indeed logically decrease the expression/phosphorylation of AMPK. siRNA and compound C are already used in these cells and in presence of shikonin but not for the other in vitro experiments (mitochondrial activity, gene expressions), which doesn’t make sense if the authors want to really state that shikonin activates AMPK. Finally: “These results suggest that shikonin directly phosphorylates AMPK.” – is confusing, as shikonin certainly does not have a kinase activity.

A1) Thank you for your comment. We totally agree with you. Unfortunately, we don’t have the siRNA-AMPK cells now. So, it is too difficult to conduct the additional experiment within revision periods. Instead of this, we analyzed mRNA expression of hepatic genes involved in mitochondrial function and fatty acid oxidation in live tissue of mice, and these results were added in the text(line 249) and figure 7.

Also, according to your suggestion, we correct the points related over-interpretations that shikonin directly regulates AMPK.

These should be clarified:

Line 178: “The shikonin induced increase in AMPK phosphorylation reduced in the presence of compound C (Fig. 1D).”

  • Line 172-174 : The results showed that shikonin treatment partly recovered AMPK phosphorylation reduced by compound C in Hepa 1-6 cells (Fig. 1D).

Line 189-195: To confirm the critical role of shikonin-induced AMPK activation, we used a specific siRNA against the AMPK gene. AMPK protein and mRNA expression levels were significantly suppressed by the AMPK siRNA compared to the control siRNA (Fig. 2). Further, knockdown of AMPK abolished shikonin mediated AMPK activation. In addition, ACC phosphorylation was dramatically decreased by siRNA transfection. The role of shikonin in AMPK activation was verified by the lack of downstream ACC activation in Hepa 1-6 cells deficient of AMPK. This data suggests that shikonin directly regulates AMPK activation.

  • We deleted Figure 2 and related part in the text to avoid confusing.
  • Line 196-199 : We performed additional experiments using siRNA against AMPK. The siRNA could efficiently block AMPK protein and mRNA expression levels. In knockdown state of AMPK, there was no change by shikonin treatment (Fig. 2). These results suggest that shikonin may be closely related to the action of AMPK.

Line 297-301: To evaluate the involvement of shikonin on direct AMPK activation, we used chemical and genetic approaches to inhibit AMPK. Compound C disrupted the shikonin-induced phosphorylation of AMPK (Fig. 1C). In a similar approach, silencing AMPK expression using siRNA transfection suppressed the shikonin-induced phosphorylation of AMPK (Fig. 2). These results suggest that shikonin directly phosphorylates AMPK.

  • Line 295-297 : To further validate our results, we examined using compound C, chemical inhibitor of AMPK. AMPK phosphorylation inhibited by compound C was partly reversed by shikonin treartment.

Q2) Same statement for in vivo experiments :

The authors should be careful about the fact that no link is shown here between AMPK and fatty acid oxidation/steatosis. The authors should reformulate some of these conclusions. For example : (please double-check the entire masnuscript)

A2) We would like to thank you for your careful read and thoughtful comments. We have corrected as you pointed out.

Line 259: Shikonin prevents high-fat diet-induced hepatic steatosis via AMPK activation

  • Line 249 : Shikonin prevents high-fat diet-induced hepatic steatosis.

Line 277-278: This study demonstrated that shikonin improves AMPK phosphorylation, increasing fatty acid oxidation in the liver, and consequently leading to a decrease in hepatic steatosis.

 Line 272-278: In the current study, we explored the therapeutic effect and its action mechanisms of shikonin on obesity induced hepatic steatosis by using OA-treated Hepa 1-6 cells and HFD-induced obese mice models. Our finding showed that shikonin could reduce the level of lipid accumulation in Hepa 1-6 cell and increase the phosphorylation levels of AMPK and ACC. Also, shikonin enhanced cellular oxygen consumption and significantly increased the fatty acid oxidation related genes such as PPARα, CPT-1, and PGC-1α. More importantly, shikonin significantly increased whole-body O2 consumption in obese mice.

 Q3) References in the text have been added or modified and helped to better understand what is already known about the effect of shikonin and derivatives. One major effect of shikonin have unfortunately been again omitted.

Line 283-284: However, the effect of shikonin on fatty liver disease had not been previously examined.

As asked previously, this should be cited and discussed:

Shikonin from Zicao prevents against non-alcoholic fatty liver disease induced by high-fat diet in rats. Weijia Yang et al., Academia Journal of Scientific Research, 2017

Shikonin alleviates hepatic fibrosis and autophagy via inhibition of TGF-β1/Smads pathway. Tong Liu et al., Journal of Gastroenterology and Hepatology, 2019

Efficacy of Acetylshikonin in Preventing Obesity and Hepatic Steatosis in db/db Mice. Mei-Ling Su et al., 2016, Molecules.

A3) We add references in revised manuscript in introduction and discussion.

  • Line 53-60 : Many studies have shown that shikonin can exert protective effects against obesity by modulating glucose tolerance, lipogenesis and β-oxidation [17-19]. Recent studies have also shown that shikonin plays a significant role on AMPK activation against adipogenesis, diabetes, hepatic carcinoma and hepatic fibrosis [20-23]. Weijia Yang et al. reported that shikonin ameliorated hepatic lipid dysregulation through PPARγ and the MMP-9/TIMP-1 axis [24]. Also, the naphthoquinone derivative of β-hydroxyisovalerylshikonin inhibited adipogenesis of 3T3-L1 cell through increased phosphorylation of AMPK and precursor SREBP-1c [25].
  • Line 282-286: Several studies have reported that shikonin can effectively decrease obesity and hepatic fibrosis [23-25]. We previously demonstrated anti-obesity effect of shikonin by inhibiting adipogenesis and lipogenesis and by increasing β-oxidation [18,19]. Also, shikonin attenuated liver fibrosis by downregulating the transforming growth factor-β1/Smads pathway and by inhibiting autophagy [26].

 Q4) As reminded above, be careful with over-interpretations and conclusions.

LKB1 is indeed an activator of AMPK. However the authors only measure the protein expression of LKB1. It cannot unfortunately be used as a indicator of LKB1 activity as LKB1 can be phosphorylated etc. Moreover, as asked on the 1st version of the reviewing – and explaining why metformin should be added as another activator of AMPK – is that shikonin (as metformin) seems to have effects on the mitochondria itself. So to temper their interpretation and add more depth to the discussion, the authors should discuss about the possible mode of action of shikonin on AMPK.

For example:

  • LKB1 cannot be excluded here
  • Shikonin could target the mitochondria and modulate the redox state of the cellLine 293-296: "Although shikonin treatment increased AMPKα phosphorylation, the LKB1 level did not change. This is notable because AMPK is typically activated through phosphorylation by the upstream kinase LKB1. Previous studies showed that These results show that shikonin activates AMPK but did not affect the upstream kinase LKB1. Therefore, shikonin may directly be responsible for the activation of AMPK."A4) We have corrected in the text as you suggested. (line 297-303)
  •  
  • In conclusion, this should be modified

Although shikonin increased AMPKα phosphorylation, the LKB1 level did not change. This is notable because AMPK is typically activated through phosphorylation by the upstream kinase LKB1 [33]. However, it has been also reported that AMPK is activated through reversible phosphorylation by LKB1 [34]. AICAR is an an adenosine analog that directly modulates AMPK with [35] or without direct activation of LKB1 [36]. Thus, it is speculated that other possible AMPK candidate may be involved in the phosphorylation of AMPK by shikonin.

Minor:

It would be really interesting, if available, to add on fig. 6:

  • weight follow-up of mice during HFD and after Shikonin treatment
  • A) According to your suggestion, we added body weight, white adipose tissue weight and morphology in Fig 5 and in the text (Line 222-228).
  •  
  • Physical activity of these mice, as increase of AMPK phosphorylation in the muscle could be linked to more activity and could potentially explain weight loss, decrease of hepatic steatosis … 
  • A) We totally agree with you. Unfortunately, we didn’t measured physical activity of mice.